# Characteristics of hospitalized patients during a large waterborne outbreak of *Campylobacter jejuni* in Norway

Nicolay Mortensen[1,2], Solveig Aalstad Jonasson[3⊙], Ingrid Viola Lavesson[4⊙], Knut Erik Emberland[5,6], Sverre Litleskare[5], Knut-Arne Wensaas[5], Guri Rortveit[6], Nina Langeland[7], Kurt Hanevik[7,8]*

**1** Children and Youth Clinic, Haukeland University Hospital, Bergen, Norway, **2** Department of Microbiology, Haukeland University Hospital, Bergen, Norway, **3** Department of Infectious Diseases, Haukeland University Hospital, Bergen, Norway, **4** Emergency Clinic, Haukeland University Hospital, Bergen, Norway, **5** Research Unit for General Practice, NORCE Norwegian Research Centre, Bergen, Norway, **6** Department of Global Public Health and Primary Care, University of Bergen, Bergen, Norway, **7** Department of Clinical Science, University of Bergen, Bergen, Norway, **8** Department of Medicine, Norwegian National Advisory Unit on Tropical Infectious Diseases, Haukeland University Hospital, Bergen, Norway

⊙ These authors contributed equally to this work.
* kurt.hanevik@uib.no

**Data Availability Statement:** All relevant data are within the manuscript and its Supporting Information files.

## Abstract

Very few reports describe all hospitalized patients with campylobacteriosis in the setting of a single waterborne outbreak. This study describes the demographics, comorbidities, clinical features, microbiology, treatment and complications of 67 hospitalized children and adults during a large waterborne outbreak of *Campylobacter jejuni* in Askoy, Norway in 2019, where more than 2000 people in a community became ill. We investigated factors that contributed to hospitalization and treatment choices. Data were collected from electronic patient records during and after the outbreak. Fifty adults and seventeen children were included with a biphasic age distribution peaking in toddlers and middle-aged adults. Most children, 14 out of 17, were below 4 years of age. Diarrhea was the most commonly reported symptom (99%), whereas few patients (9%) reported bloody stools. Comorbidities were frequent in adults (63%) and included cardiovascular disease, pre-existing gastrointestinal disease or chronic renal failure. Comorbidities in children (47%) were dominated by pulmonary and gastrointestinal diseases. Adult patients appeared more severely ill than children with longer duration of stay, higher levels of serum creatinine and CRP and rehydration therapy. Ninety-two percent of adult patients were treated with intravenous fluid as compared with 12% of children. Almost half of the admitted children received antibiotics. Two patients died, including a toddler. Both had significant complicating factors. The demographic and clinical findings presented may be useful for health care planning and patient management in *Campylobacter* outbreaks both in primary health care and in hospitals.

**Funding:** NL received a grant from Helse Vest (https://helse-vest.no/, number F-10626).

**Competing interests:** The authors have declared that no competing interests exist.

## Introduction

*Campylobacter* is the most frequent bacterial cause of gastroenteritis in both high- and low-income countries [1], and outbreaks are often water-borne [2]. When public water supply systems are the source of infection, large parts of a population can potentially be affected in a short period. The infection is usually self-limiting with no need for antibiotic treatment, and mortality is low [3]. Associated complications include Guillian-Barré syndrome, reactive arthritis, and myocarditis [4–6].

An outbreak of infectious gastroenteritis was suspected in the Askoøy municipality in the Western part of Norway on June 6th after a period of drought followed by heavy rainfall [7]. An unusually high number of contacts due to gastroenteritis was made to the local primary health care services. The following day an elevated water reservoir was shown to be the source. A boil water advisory was issued, and the reservoir was disconnected from the water supply. Being the primary hospital for Askoy, Haukeland University Hospital experienced an increased number of admissions due to infectious gastroenteritis. Testing of patient fecal samples found *Campylobacter jejuni* to be the etiologic agent. Investigations by the Norwegian Institute of Public Health, Norwegian Food Safety Authority and Askoy Municipality found the outbreak to be clonal as *Campylobacter jejuni* isolates from patients and water samples were identical by whole-genome sequencing. The specific strain of *Campylobacter jejuni* (sequence type ST1701) did not have any close genetic relationship to other publicly available strains, although poultry or birds were identified as a possible source. Contamination occurred by leaching of animal feces through cracks in the mountain into the water reservoir between May 30th and June 3rd. It is estimated that at least 2000 persons developed symptoms of gastroenteritis [7]. Judging by the number of people affected, this was the largest documented single pathogen outbreak of *Campylobacter jejuni* in the world since the Klarup (Denmark) outbreak in 1995–96 [8].

Existing data on hospitalized patients during *Campylobacter* outbreaks are scarce, and it was hypothesized that important knowledge regarding comorbidity of hospitalized patients and factors affecting the hospitalization and treatment choices could be gained from this large outbreak. The aim of this study was to describe the demographic, clinical and microbiological characteristics of patients admitted to hospital during the outbreak.

## Material and methods

### Study setting and design

Askoy is an island municipality outside of Bergen, Norway, with a population of 29 553 at the time of the outbreak. Haukeland University Hospital is the island's primary hospital.

We conducted an observational study of patients admitted to hospital during the outbreak between May 31st and June 18th. The study population was children and adults living or working in Askoy during the outbreak, and who were admitted to hospital with signs of gastrointestinal infection.

### Inclusion and data collection

Patients were identified by searching the hospital admission registry for ICD-10 codes related to infectious gastrointestinal symptoms, *Campylobacter* infection or consumption of contaminated water (A04.5, A04.8, A04.9, A08.3, A08.4, A08.5, A09.0 and Z58.2). For patients with home address outside of Askoy a code for *Campylobacter*-infection (A04.5) was mandatory. In addition, we identified individuals from Askoy not included in the search who in the study

period either were registered in the Emergency Department admissions database with diarrhea as the cause of admission or had a fecal test positive for *Campylobacter jejuni*.

Three of the authors (NM, VL and SJ) reviewed the electronic patient records. We defined a case as a patient fulfilling three of five of the following criteria, including at least one of the two first; 1. Home address in area with water supply from the contaminated water reservoir, 2. having consumed water from the contaminated water reservoir, 3. diarrhea, 4. fever, 5. abdominal pain. We extracted demographic and admission related variables (age, sex, comorbidities, date of admission, length of stay, cause for admission, readmission), symptoms (fever, dehydration, diarrhea, vomiting, abdominal-, head-, chest- or joint pain, sleeping difficulties, bloody stools), clinical parameters (heart rate, respiratory rate, blood pressure, body temperature, SpO2), blood tests (C-Reactive Protein (CRP, including results in admission letters), leukocyte count, hemoglobin concentration, creatinine), treatment (antibiotics, fluids, pain relief), microbiological sample results (fecal test, blood cultures) and complications (death, readmissions, other). A patient was registered as dehydrated based on the clinical assessment and conclusion by the attending doctor. For all blood samples the value most deviant from the appropriate reference range during the first 24 hours after admission was registered. A child was defined as a person younger than sixteen years. We categorized comorbidities as pulmonary-, cardiovascular-, rheumatological-, psychiatric-, gastrointestinal- and neurological disease, cancer, diabetes and chronic renal failure. A single patient could be registered with multiple comorbidities.

Severity of illness was assessed by SIRS (Systemic Inflammatory Response Syndrome) in adults and children with age appropriate reference ranges for parameters) and qSOFA (quick Sepsis Organ Failure Assessment) scores (only adults). SIRS is an extensive inflammatory state in response to an infectious or noninfectious insult. It consists of four criteria in which two criteria or more are needed for a 'positive' score. In cases with a suspected infection, SIRS was widely used in hospitals as a screening for sepsis. qSOFA is a combination of clinical criteria used to identify adult patients with suspected infection who have greater risk of poor outcome. Two criteria or more is considered a 'positive' score and indicates higher mortality risk. By international consensus qSOFA has replaced SIRS [9].

Fisher exact test was used for comparisons of symptoms and treatment between adults and children, and MannWhitney two-tailed non-parametric test was used for continuous variables. Significance level was p<0.05. SPSS version 25 was used for statistics. Excel 2016 version 16.0.5110.1000 was used for graphics.

### Ethics

Two adults and the parents of one child withdrew consent and were excluded. All included patients, or parents in case of children under the age of 16, gave active or passive consent to participate. The Regional Ethics Committee for Medical and Health Research Ethics (REC) approved the study and consent procedures.

### Results

A total of 67 patients fulfilled the case definition and were included in the study. Of these, 50 were adults and 17 children. All children were younger than nine years, and 14 were infants and toddlers under the age of four years (Fig 1).

Patient characteristics are shown in Table 1. The mean length of hospital stay for adults was 2 days, with a maximum of 9 days. Maximum length of stay for children was 2 days. Ten children stayed for less than 24 hours, and of these 50% (5/10) were discharged directly from the pediatric emergency department after an initial assessment and observation by the pediatrician

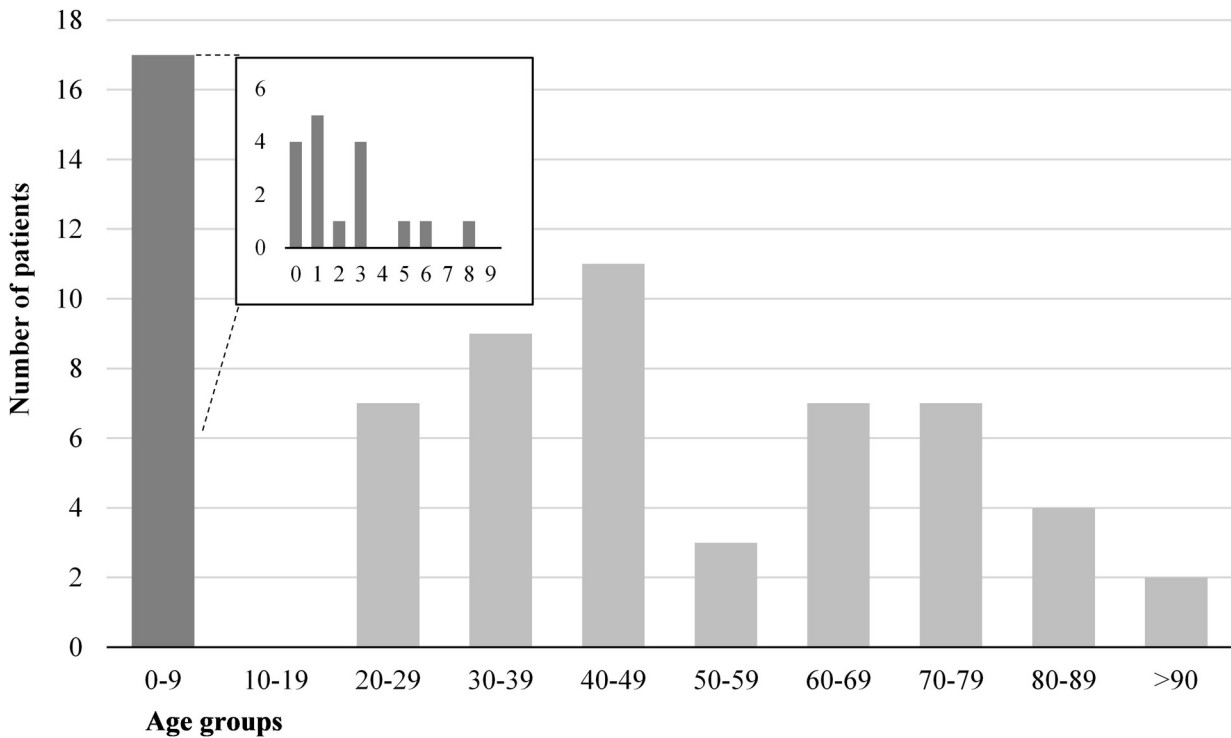

**Fig 1. Number of patients with campylobacteriosis by age group, in an outbreak in Askoy, Norway 2019.** The bar for 0–9 years is further divided in a separate diagram by one-year intervals.

on call. Comorbidity was registered in 68% of adults and 47% of children. Cardiovascular disease (30%), gastrointestinal disease (28%) and chronic renal failure (22%) were major comorbid conditions among adults. In children, the major comorbidities were chronic gastrointestinal disease and pulmonary disease, which were equally prevalent (29%). Among 5 children with pulmonary disease, 4 had asthma.

**Table 1. Characteristics among hospitalized patients with campylobacteriosis during an outbreak in Askoy, Norway 2019, including comorbidity categorized into disease groups.**

|  | Adults (≥16) | | Children (<16) | | Total | |
|---|---|---|---|---|---|---|
| Total, N (% of all patients) | 50 | (75) | 17 | (25) | 67 | (100) |
| Female, n (% total) | 25 | (50) | 7 | (41) | 32 | (48) |
| Age, years, mean (range) | 52.3 | (20–93.2) | 2.6 | (0.2–8.7) | 39.7 | (0.2–93.2) |
| Length of stay, days, mean (range) | 2 | (0–9) | 0 | (0–2) | 1.5 | (0–9) |
| Comorbidity, n (% total) | 34 | (68) | 8 | (47) | 42 | (63) |
| Cancer | 1 | (2) | 0 | (0) | 1 | (1) |
| Pulmonary disease | 6 | (12) | 5 | (29) | 11 | (16) |
| Cardiovascular disease | 15 | (30) | 1 | (6) | 16 | (24) |
| Diabetes | 3 | (6) | 0 | (0) | 3 | (4) |
| Rheumatological disease | 5 | (10) | 0 | (0) | 5 | (7) |
| Psychiatric disease | 8 | (16) | 0 | (0) | 8 | (12) |
| Gastrointestinal disease | 14 | (28) | 5 | (29) | 19 | (28) |
| Neurological disease | 6 | (12) | 1 | (6) | 7 | (10) |
| Chronic renal failure | 11 | (22) | 0 | (0) | 11 | (16) |

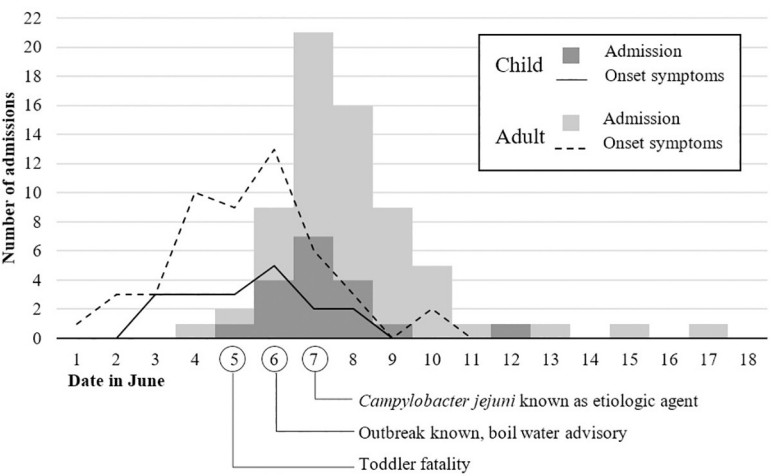

**Fig 2. Total number of admissions for children and adults by date in June 2019 in an outbreak of campylobacteriosis in Askoy, Norway.** Onset of symptoms for both groups represented by lines. Number of patients on y-axis is for both admission date and date of symptom onset.

Fig 2 shows number of admissions and onset of symptoms by date in June 2019. Mean duration of symptoms before being admitted was slightly longer in adults (three days) than in children (two days). The increase in hospital admission due to gastrointestinal infection came around the same time as a toddler with diarrheal disease died. The number of admissions declined between two to three days after the boil water advisory was issued and the suspected water reservoir was disconnected from the water supply.

Symptoms presented at admission are shown in Table 2. All patients except for one adult reported diarrhea (66/67, 99%). Only 6 patients reported bloody stools, but significantly more adults than children were evaluated to be dehydrated. Vomiting and fever were significantly more common in children. Symptoms dependent on subjective communication such as abdominal pain were omitted for children, as they were too young to reliably report these. 18% of adults reported joint pain.

**Table 2. Symptoms among hospitalized children and adults in an outbreak of campylobacteriosis in Askoy, Norway 2019.**

| | Total | | Adults (≥16) | | Children (<16) | | |
|---|---|---|---|---|---|---|---|
| | n | % total | n | % total | n | % total | OR (95% CI) |
| Diarrhea | 66 | 99% | 49 | 98% | 17 | 100% | ns |
| Bloody stools | 6 | 9% | 4 | 8% | 2 | 12% | ns |
| Dehydration | 45 | 67% | 40 | 80% | 5 | 29% | 9.60 (2.74–33.6) |
| Fever[a] | 27 | 40% | 13 | 26% | 14 | 82% | 0.08 (0.02–0.31) |
| Vomiting | 25 | 37% | 13 | 26% | 12 | 71% | 0.15 (0.04–0.50) |
| Abdominal pain | na | - | 35 | 70% | na | - | na |
| Headache | na | - | 9 | 18% | na | - | na |
| Joint pain | na | - | 9 | 18% | na | - | na |
| Insomnia | na | - | 6 | 12% | na | - | na |
| Chest pain | na | - | 3 | 6% | na | - | na |

[a] Fever was either self-reported or measured as temperature ≥38 degrees centigrade at admission.

ns = not significant, na = not applicable.

**Table 3. Treatment of hospitalized children and adults in an outbreak of campylobacteriosis in Askoy, Norway 2019.**

|  | Total | | Adults (≥16) | | Children (<16) | |  |
|---|---|---|---|---|---|---|---|
|  | n | % total | n | % total | n | % total | OR (95% CI) |
| Intravenous fluid | 48 | *72%* | 46 | *92%* | 2 | *12%* | 86.3 (14.3–519.0) |
| Fluid by feeding tube | 2 | *3%* | 0 | *0%* | 2 | *12%* | na* |
| Antibiotic treatment started, any indication | 21 | *31%* | 12 | *24%* | 9 | *53%* | 0.28 (0.09–0.89) |
| Antibiotic treatment for *Campylobacter* started | 15 | *22%* | 7 | *14%* | 8 | *47%* | 0.18 (0.05–0.64) |
| Analgesics other than acetaminophen/paracetamol | 19 | *28%* | 18 | *36%* | 1 | *6%* | 9.00 (1.1–73.6) |

na = not applicable

* Fluid by feeding tube is very rarely considered in adults.

CRP results was available from 15 children, and results of additional blood samples were obtained from 13 children. Every adult patient had blood samples taken. Adults had mean CRP levels of 112mg/L (range 6–302), which was higher than children, who had a mean level of 43mg/L (range 0–149, $p < 0.001$). Serum creatinine levels were above reference range in 14 male adults (mean 115 umol/L, range 66–179, reference range 60–105) and in 1 female adult (mean 69 umol/L, range 64–100, reference range 45–90). The mean serum creatinine in the nine children where it was available was 28 umol/L (range 18–56, reference range 20–70). Seven of fourteen (41%) male patients without chronic renal failure had serum creatinine levels above the reference range, while all women and children were within reference ranges.

Thirty percent of adult patients (15/50) had hemoglobin concentrations below age and sex specific reference ranges compared to 14% (2/13) of the children. Among 15 adults with low hemoglobin concentration, 2 (13%) had bloody stools, and 11 (73%) had comorbid conditions. All 4 adult patients with bloody stools also had comorbid conditions as opposed to children, where no comorbidities were observed in this group.

Regarding treatment, the majority of adult patients (92%) received intravenous fluids, compared to 12% of the children (Table 3). Nearly half of the children (8/17, 47%) received antibiotic treatment for *Campylobacter*-infection. Erythromycin was used in seven children and azithromycin in one. One child was treated for septic shock of unknown origin. The use of antibiotics for *Campylobacter*-infection was significantly lower in adult patients (7/50, 14%), where four patients were given erythromycin and three azithromycin. Among adults there was a tendency toward an increasing proportion of antibiotic use with age; of 13 patients aged seventy years or older, four patients received antibiotic treatment for *Campylobacter*. Five adult patients received antibiotics against other infections, including UTI and cholecystitis, none for sepsis.

Fifty-nine patients had fecal *Campylobacter* tests taken (94%, 47/50 of adult patients and 71%, 12/17 of children). All were positive for *Campylobacter jejuni*, either by growth in culture (90%, 53/59), by positive PCR (92%, 54/59), or both. In addition, 14% (8/59) were PCR positive for other possible infectious agents (three adults were positive for the *eae* gene of enteropathogenic *Escherichia coli*, one adult and two children were positive for adenovirus, one adult positive for norovirus and one child positive for astrovirus). No cultures showed growth of pathogenic bacteria other than *Campylobacter jejuni*. All blood cultures were negative for pathogenic bacteria, including *Campylobacter jejuni*.

No adult patients had a positive qSOFA score. Approximately 40% of both children (7/17) and adult (21/50) patients had a SIRS score ≥ 2 criteria. Tachycardia (above the age appropriate reference range) was more common in adults (56%, 28/50) than in children (12%, 2/17, p = 0.002), and vice versa for respiratory rate (50%, 25/50 versus 88%, 15/17, p = 0.009). Fever

($\geq$ 38.5 degrees centigrade) was present in 36% (18/50) of adults and 41% (7/17) of children. There was no apparent correlation between initiation of antibiotics and SIRS or qSOFA criteria. Three adult patients had SpO2 between 90–94% (reference $\geq$95%) at admission, all with previous cardiovascular disease but no pulmonary disease. One child had SpO2 below 90% (reference $\geq$95%).

Complications were fatal as one child and one adult died. The adult patient was severely immunocompromised due to treatment for an underlying condition. The child died of sepsis caused by *Streptococcus pyogenes* with concomitant *Campylobacter* gastroenteritis. One adult suffered from peri-myocarditis in the course of the infection. Two adults and three children were readmitted for persisting diarrhea or bloody stools within two weeks of discharge.

## Discussion

Several clinical reports on waterborne outbreaks of *Campylobacter* have been published over the previous three decades, including many in the Nordic countries [8, 10–14]. Clinical features of patients with campylobacteriosis have also been well documented [15–19], including among pediatric patients [15, 20–22]. However, very few studies with a clinical focus on hospitalized patients in the setting of an outbreak have been published. The present study is the largest published on hospitalized patients from a single clonal outbreak, and information on clinical characteristics of this patient group could be useful for emergency room clinicians.

The demographic curve was biphasic and showed a typical pattern where infants, toddlers and middle-aged adults were represented. Distribution of age and sex in hospitalized patients were similar to those of contacts made to the local primary health care services during the outbreak [7], except that infants and toddlers dominated among children, and no adolescents were admitted. Several reports [17, 18, 23] have found similar demographic profiles of patients affected by campylobacteriosis, although a higher proportion of young adults tend to be admitted than in our cohort [24, 25]. Demographic data from Askøy shows that persons in their twenties are less numerous than both younger and older age groups [26]. Since Askoy is a relatively small municipality, one possible explanation might be that persons in this age group gravitate towards the neighboring city of Bergen for its educational and cultural institutions. However, the contacts made to the Askoy primary health care services during the outbreak were found to be evenly distributed among age groups [7].

Comorbidities were common in both adults and children. Based on data from Statistics Norway, showing that 1% of the adult ($\geq$16 years) population suffer from chronic renal failure, patients admitted with this comorbidity were over-represented in our cohort (22%). The prevalence of epilepsy and constipation among children in our patient material was similar to that found in national data, suggesting that these conditions did not increase the risk for hospitalization during the outbreak [27].

Symptoms were typical for campylobacteriosis, with some differences between adults and children. All patients except one adult reported diarrhea. A Norwegian study of sporadic cases of campylobacteriosis [19] found that 98.5% of patients reported diarrhea, which is in line with our findings. For some symptoms, like abdominal pain, headache, joint pain and chest pain, differences between children and adults could be due to difficulties for young children expressing these complaints as something other than general discomfort. A large study on 662 hospitalized, culture-confirmed patients with campylobacteriosis from USA [25] found that 37% of patients had vomiting, which is similar to our cohort (31%). The percentages of patients with abdominal pain (70%, only adults), fever (74%) and bloody diarrhea (39%) were generally higher than in our material, particularly bloody diarrhea which was low (9%) in the present study. Bloody diarrhea is widely regarded as a common feature of *Campylobacter*

enteritis, and prevalences range from 39% to over 90% [25, 28]. Virulence factors of the particular strain of *Campylobacter jejuni* is associated with the presence of bloody stools [29], and further genetic studies of the Askøy isolate could potentially reveal the presence of such factors. Other differences in prevalence of symptoms between studies may have several explanations. Reporting methods may vary and a higher threshold for seeking health care could affect severity of illness, and consequently the rate of symptoms, of those hospitalized in Norway and other countries.

There were differences in both dehydration frequencies and treatment strategies between adults and children. More adults than children were dehydrated, had elevated levels of serum creatinine and nearly all adults received intravenous fluids. Since no standardized tool is used systematically among doctors to evaluate if a patient is dehydrated or not, comparison among patients is difficult. Still, it is striking that 92% of adults received intravenous fluids. The reason for this could in part be due to local practices and accustomed routine in the adult Emergency Department, where ordering of blood tests and placement of intravenous cannula is often carried out by nurses as a routine prior or parallel to the clinical assessment by a medical doctor. Given the potentially painful and frightening nature of such procedures on children, pediatricians will more often critically consider the benefit for each patient individually. Fluids need to be ordered according to the child's weight and degree of dehydration and hence cannot be initiated without the doctor's involvement. For those who are not severely dehydrated or have an ongoing significant gastrointestinal loss of fluids due to diarrhea or vomiting, an initial attempt at dehydration orally or via feeding tube is often given. In this outbreak, five of ten children who stayed for less than 24 hours were discharged with their caretakers after careful medical assessment and information on how to continue treatment and monitor the condition at home. This strategy could likely be considered more often in the adult Emergency Department, particularly in a setting of a large number of referrals over a short period.

Campylobacteriosis is usually a self-limiting infection. National and international guidelines generally recommend antimicrobial treatment only in invasive cases or patients with risk of severe disease [30, 31]. Nearly half of the children were started on antibiotics for campylobacteriosis upon admission. This is a much higher number than expected in Norway, and particularly when compared to adults of whom approximately one in ten was started on antibiotics despite generally appearing more affected by the disease and having more comorbidity. Norway in general has a rather cautious policy regarding use of antibiotics as compared to other countries [32]. A previous study of sporadic campylobacteriosis in Norway report antibiotic treatment rates of 16% [19]. The study did not differentiate between hospitalized patients and outpatients. A more current study from USA reported treatment rates that were consistent among age groups and hospitalization status, at 35% [25]. Data from the Askoy primary health care services show that approximately 1% of patients who made contact during the current outbreak were prescribed antibiotic treatment. Around the same time as the outbreak was known among health care workers and the population of Askoy, it was discovered that a toddler who died unexpectedly had tested positive for *Campylobacter jejuni*. Media coverage was extensive. This may have influenced health care seeking behavior and lowered the threshold for primary care doctors to refer children to the hospital.

There are limitations to our study. Even though this was a large outbreak, the number of hospitalized patients was small and statistical analyses were done only when deemed feasible. Clinical data was registered prospectively in electronic patient records by doctors on call, but were extracted retrospectively by the authors. This design may have caused some bias in the recording of symptoms or comorbid conditions as no standardized questionnaires other than what is clinical practice at our hospital was employed. Still, larger cohorts of hospitalized patients during an outbreak of *Campylobacter* infection will be rare in high-income countries,

implying that the insights from the current study is useful for health care planning and patient management both in primary health care and in hospitals.

## Conclusions

Among hospitalized patients during this *Campylobacter* outbreak, adults generally appeared more severely ill than children, as judged by extent of treatment, length of stay, level of serum creatinine and CRP. Adults with chronic renal failure were over-represented. However, fever and vomiting were seen more often in children than adults. Bloody stool was a less prevalent feature than previously reported for campylobacteriosis. Nearly every adult patient was rehydrated by intravenous fluids regardless of apparent degree of dehydration, while oral rehydration was more commonly considered in children. More children were started on treatment with antibiotics than could be expected from national and international guidelines. Media attention around a fatal outcome during the initial phase of an outbreak may have affected the threshold for both referral and clinical decision making in the emergency room.

## Supporting information

**S1 Dataset. Data underlying the results described in this study.** Data on individual participants are anonymized in compliance with GDPR requirements. Categories of patients within a given variable with less than 5 patients are merged or categorized to larger groups (such as age groups). Some variables impossible to categorize had to be removed from the dataset. Variable labels are included in a separate sheet in the Excel file.
(XLSX)

## Acknowledgments

Thanks to study nurses at the Department of Infectious Diseases and the Research department at Haukeland University Hospital for initial coordination.

## Author Contributions

**Conceptualization:** Nicolay Mortensen, Solveig Aalstad Jonasson, Ingrid Viola Lavesson, Knut Erik Emberland, Sverre Litleskare, Knut-Arne Wensaas, Guri Rortveit, Nina Langeland, Kurt Hanevik.

**Formal analysis:** Nicolay Mortensen, Solveig Aalstad Jonasson, Ingrid Viola Lavesson, Kurt Hanevik.

**Funding acquisition:** Guri Rortveit, Nina Langeland, Kurt Hanevik.

**Investigation:** Nicolay Mortensen, Solveig Aalstad Jonasson, Ingrid Viola Lavesson, Nina Langeland, Kurt Hanevik.

**Methodology:** Nicolay Mortensen, Solveig Aalstad Jonasson, Ingrid Viola Lavesson, Sverre Litleskare, Knut-Arne Wensaas, Guri Rortveit, Nina Langeland, Kurt Hanevik.

**Project administration:** Guri Rortveit, Nina Langeland, Kurt Hanevik.

**Resources:** Solveig Aalstad Jonasson, Ingrid Viola Lavesson, Knut Erik Emberland, Kurt Hanevik.

**Visualization:** Nicolay Mortensen, Nina Langeland, Kurt Hanevik.

**Writing – original draft:** Nicolay Mortensen.

**Writing – review & editing:** Solveig Aalstad Jonasson, Ingrid Viola Lavesson, Knut Erik Emberland, Sverre Litleskare, Knut-Arne Wensaas, Guri Rortveit, Nina Langeland, Kurt Hanevik.

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
