## [Decision Letter · Decision Letter 0]

18 Jan 2021

PONE-D-20-40362

Characteristics of hospitalized patients during a large waterborne outbreak of Campylobacter jejuni in Norway

PLOS ONE

Dear Dr. Mortensen,

Thank you for submitting your manuscript to PLOS ONE. After careful consideration, we feel that it has merit but does not fully meet PLOS ONE’s publication criteria as it currently stands. Therefore, we invite you to submit a revised version of the manuscript that addresses the points raised during the review process.

We look forward to receiving your revised manuscript.

Kind regards,

Tai-Heng Chen, M.D.

Academic Editor

PLOS ONE

2. For more information on PLOS ONE's expectations for statistical reporting, please see https://journals.plos.org/plosone/s/submission-guidelines.#loc-statistical-reporting

Please update your Methods and Results sections accordingly.

3. You indicated that you had ethical approval for your study.

In your Methods section, please ensure you have also stated whether you obtained consent from parents or guardians of the minors (<18) included in the study or whether the research ethics committee or IRB specifically waived the need for their consent.

Reviewers' comments:

Reviewer's Responses to Questions

**Comments to the Author**

1. Is the manuscript technically sound, and do the data support the conclusions?

Reviewer #1: Yes

Reviewer #2: Yes

Reviewer #3: Yes

2. Has the statistical analysis been performed appropriately and rigorously? 

Reviewer #1: Yes

Reviewer #2: Yes

Reviewer #3: Yes

3. Have the authors made all data underlying the findings in their manuscript fully available?

Reviewer #1: Yes

Reviewer #2: Yes

Reviewer #3: Yes

4. Is the manuscript presented in an intelligible fashion and written in standard English?

Reviewer #1: Yes

Reviewer #2: Yes

Reviewer #3: Yes

5. Review Comments to the Author

Reviewer #1: This paper is well done and interesting. I have a couple of minor suggestions; the paper would benefit from grammatical editing, but these changes are relatively minor. The tables would be improved if the top headers (Total, Adult, Children) were centered over the two columns below. Otherwise, this is an excellent paper and worth publishing.

Reviewer #2: This paper presents the clinical characteristics of hospitalized patienst suffering of campylobacteriosis during a large water-borne outbreak in Norway. The authors included 67 patients in their study.

General comments:

The manuscript is well written and concise.

I understand that the patient included in the study were affected by the Alskoy outbreak previously described in Hyllestad et al. 2020. It would be useful for the reader to add more useful details about the outbreak in the introduction, in particular about the strains that caused the outbreak.

Related to my previous point, the authors noticed lower prevalence of bloody diarrhea in their patent cohort and suggest that this might be related the genetic of the outbreak strain. Since the strains isolated during this outbreak, it might be possible for the authors to substantiate their claim by determining the presence of the associated genes in the sequenced genome. However, it seems that these genomes have not been deposited by Hyllestad et al. in appropriate database, so it might not be trivial to do this analysis.

Specific comments:

The y-axis of figure 2 is confusing since two dataset are presented: onset of symptoms and admission date. I guess the y-axis is also the number of people with onset of symptoms at a particular day of the outbreak? This should be clarified.

Please define CRP.

Reviewer #3: This paper provides a detailed description of hospitalized patients from a waterborne Campy outbreak in Norway using electronic healthcare records. The authors assess factors related to hospitalization and treatments in children and adults.

Minor edits

-line 67: typo “Judged” should be “Judging”

-table 1: in row “Length of stay”, there is a typo “1,5”

-table 3: in the last row, part of the OR appears to be deleted in the pdf I received

-line 215: typo “38,5”

Methods

-lines 89-124: excellent description of the data collected and definitions used

Results

-figures are very nicely made

-Table 1: consider adding the ages for adults/children under each heading i.e. Adults (17+) Children (<17) just as a reminder for readers

-Table 1: because you use (% total) to refer to % of each category for most of the table (for Female and Comorbidity categories), it is a little confusing to use if for the first row (Total) where you mean % of all patients. I suggest you change the first row label to “Total, N (% of all patients)”. Also consider changing the style this table to the same as tables 2 and 3 (i.e. separate columns for n and %)

-lines 158-163: It might be helpful to include the incubation time for Campy in this section

-lines 208-209: Do you have any additional information on what this PCR was detecting?

Discussion

-line 244: Do you have any insights into why this might be? Are young adults less likely to seek healthcare or do you think they have less severe symptoms?

-lines 303-304: Is this 1% among all patients for all primary healthcare services? Or just infectious disease patients?

Overall comments

This is a well-written paper and with a comprehensive and detailed clinical dataset. The information provided in the paper will be useful for clinicians or public health representatives in outbreak situations. The data additionally have potential for use in other studies or to answer additional research questions.

I have no major concerns, only suggestions for edits and clarifications in the text.

6. PLOS authors have the option to publish the peer review history of their article (what does this mean?). If published, this will include your full peer review and any attached files.

Reviewer #1: No

Reviewer #2: No

Reviewer #3: **Yes: **Rachel Sippy

---

## [Author Response · Author response to Decision Letter 0]

10 Feb 2021

Response to reviewers

Manuscript: Mortensen et al. Characteristics of hospitalized patients during a large waterborne outbreak of Campylobacter jejuni in Norway

Please find our responses marked by an asterix* below each comment from editor and reviewers. All changes are all visible in the file “Revised Manuscript with Track Changes”.

Reviewer #1: 

This paper is well done and interesting. I have a couple of minor suggestions; the paper would benefit from grammatical editing, but these changes are relatively minor. The tables would be improved if the top headers (Total, Adult, Children) were centered over the two columns below. Otherwise, this is an excellent paper and worth publishing.

*We thank the reviewer for the kind remarks on our paper. Tables have been formatted so that the top headers center over their respective columns.

Reviewer #2: 

This paper presents the clinical characteristics of hospitalized patients suffering of campylobacteriosis during a large water-borne outbreak in Norway. The authors included 67 patients in their study.

General comments:

The manuscript is well written and concise.

*We thank the reviewer for these kind remarks.

I understand that the patient included in the study were affected by the Askoy outbreak previously described in Hyllestad et al. 2020. It would be useful for the reader to add more useful details about the outbreak in the introduction, in particular about the strains that caused the outbreak.

*We have in the revised manuscript introduction supplied more details on the outbreak itself with an emphasis on the outbreak strain of C. jejuni (lines 62-68) 

Related to my previous point, the authors noticed lower prevalence of bloody diarrhea in their patent cohort and suggest that this might be related the genetic of the outbreak strain. Since the strains isolated during this outbreak, it might be possible for the authors to substantiate their claim by determining the presence of the associated genes in the sequenced genome. However, it seems that these genomes have not been deposited by Hyllestad et al. in appropriate database, so it might not be trivial to do this analysis.

*We agree that a possible correlation between the presence of genes coding for specific virulence factors and our clinical data would be most interesting to explore. The C. jejuni isolates from patients in our study may become available for whole genome sequencing at a later stage.

Specific comments:

The y-axis of figure 2 is confusing since two dataset are presented: onset of symptoms and admission date. I guess the y-axis is also the number of people with onset of symptoms at a particular day of the outbreak? This should be clarified.

*We agree with the reviewer and have changed the figure 2 text has been changed to better describe the y-axis.

Please define CRP.

*CRP has been defined (line 107).

Reviewer #3

This paper provides a detailed description of hospitalized patients from a waterborne Campy outbreak in Norway using electronic healthcare records. The authors assess factors related to hospitalization and treatments in children and adults.

Minor edits

-line 67: typo “Judged” should be “Judging”

-table 1: in row “Length of stay”, there is a typo “1,5”

-table 3: in the last row, part of the OR appears to be deleted in the pdf I received

-line 215: typo “38,5”

*Thank you for pointing this out. Typos have been corrected and the p-value that was missing is added.

Methods

-lines 89-124: excellent description of the data collected and definitions used

Results

- figures are very nicely made

-Table 1: consider adding the ages for adults/children under each heading i.e. Adults (17+) Children (<17) just as a reminder for readers

*This is a good suggestion. The table has been corrected accordingly. Note also that we have specified the definition of a child in the text (line 113). It originally said that a patient was considered a child if sixteen years or younger. It should say below sixteen years, and this has been corrected. No included patients are around this age. All statistics have been performed based on the intended definition.

-Table 1: because you use (% total) to refer to % of each category for most of the table (for Female and Comorbidity categories), it is a little confusing to use if for the first row (Total) where you mean % of all patients. I suggest you change the first row label to “Total, N (% of all patients)”. Also consider changing the style this table to the same as tables 2 and 3 (i.e. separate columns for n and %)

*We have changed the first row label as suggested. As the variables in table 1 have different parameters (mean, range, %, etc.) for each row the style with separate columns for “n” and “%” used in tables 2 and 3 has not been applied for table 1.

-lines 158-163: It might be helpful to include the incubation time for Campy in this section

*We agree that the incubation time would be helpful. Since we unfortunately do not know the exact date of exposure for each patient we have not been able to calculate incubation time.

-lines 208-209: Do you have any additional information on what this PCR was detecting?

Information on additional PCR results have been added to the text (lines 208-214).

Discussion

-line 244: Do you have any insights into why this might be? Are young adults less likely to seek healthcare or do you think they have less severe symptoms?

*Thank you for interesting questions. It appears that the reason for why young adults tend to be over-represented in outbreaks of campylobacteriosis is not well understood among researchers. Also, there is a difference in age distribution for campylobacteriosis between high- and low-income countries, possibly due to degree of exposure during childhood and acquired immunity. As to why the age distribution in our cohort differed from what is most often reported in high income countries we hypothesize that part of the reason could be the demographics in Askøy municipality where young adults in their twenties are less numerous compared to other age groips. We have added this to the discussion, including a reference to Statistics Norway that shows the age distribution of Askøy inhabitants https://www.ssb.no/kommunefakta/askoy (lines 250-255).

-lines 303-304: Is this 1% among all patients for all primary healthcare services? Or just infectious disease patients?

*The 1% is among all patients who made contact to the Askøy primary health care services during the initial phase of the outbreak. This has now been clarified in the text (lines 314-315).

Overall comments

This is a well-written paper and with a comprehensive and detailed clinical dataset. The information provided in the paper will be useful for clinicians or public health representatives in outbreak situations. The data additionally have potential for use in other studies or to answer additional research questions.

I have no major concerns, only suggestions for edits and clarifications in the text.

*We thank the reviewer for kind and constructive remarks.

---

## [Decision Letter · Decision Letter 1]

1 Mar 2021

Characteristics of hospitalized patients during a large waterborne outbreak of Campylobacter jejuni in Norway

PONE-D-20-40362R1

Dear Dr. Mortensen,

We’re pleased to inform you that your manuscript has been judged scientifically suitable for publication and will be formally accepted for publication once it meets all outstanding technical requirements.

Kind regards,

Tai-Heng Chen, M.D.

Academic Editor

PLOS ONE

Reviewers' comments:

Reviewer's Responses to Questions

**Comments to the Author**

1. If the authors have adequately addressed your comments raised in a previous round of review and you feel that this manuscript is now acceptable for publication, you may indicate that here to bypass the “Comments to the Author” section, enter your conflict of interest statement in the “Confidential to Editor” section, and submit your "Accept" recommendation.

Reviewer #1: All comments have been addressed

Reviewer #2: All comments have been addressed

2. Is the manuscript technically sound, and do the data support the conclusions?

Reviewer #1: Yes

Reviewer #2: Yes

3. Has the statistical analysis been performed appropriately and rigorously? 

Reviewer #1: N/A

Reviewer #2: Yes

4. Have the authors made all data underlying the findings in their manuscript fully available?

Reviewer #1: Yes

Reviewer #2: Yes

5. Is the manuscript presented in an intelligible fashion and written in standard English?

Reviewer #1: Yes

Reviewer #2: Yes

6. Review Comments to the Author

Reviewer #1: (No Response)

Reviewer #2: (No Response)

7. PLOS authors have the option to publish the peer review history of their article (what does this mean?). If published, this will include your full peer review and any attached files.

Reviewer #1: No

Reviewer #2: No

---

## [Editor Report · Acceptance letter]

12 Mar 2021

PONE-D-20-40362R1 

Characteristics of hospitalized patients during a large waterborne outbreak of *Campylobacter jejuni* in Norway 

Dear Dr. Hanevik:

I'm pleased to inform you that your manuscript has been deemed suitable for publication in PLOS ONE. Congratulations! Your manuscript is now with our production department. 

Kind regards, 

on behalf of

Dr. Tai-Heng Chen 

Academic Editor

PLOS ONE